# The SUMOylation Pathway Components Are Required for Vegetative Growth, Asexual Development, Cytotoxic Responses, and Programmed Cell Death Events in *Fusarium oxysporum* f. sp. *niveum*

**DOI:** 10.3390/jof9010094

**Published:** 2023-01-09

**Authors:** Muhammad Noman, Yizhou Gao, Hui Wang, Xiaohui Xiong, Jiajing Wang, Dayong Li, Fengming Song

**Affiliations:** 1Ministry of Agriculture Key Laboratory of Molecular Biology of Crop Pathogens and Insect Pests, Institute of Biotechnology, Zhejiang University, Hangzhou 310058, China; 2Key Laboratory of Biology of Crop Pathogens and Insects of Zhejiang Province, Institute of Biotechnology, Zhejiang University, Hangzhou 310058, China

**Keywords:** asexual reproduction, cell death, cell wall integrity, *Fusarium oxysporum* f. sp. *niveum*, mycelial growth, stress response, SUMOylation pathway, watermelon

## Abstract

SUMOylation is an essential protein modification process that regulates numerous crucial cellular and biochemical processes in phytopathogenic fungi, and thus plays important roles in multiple biological functions. The present study characterizes the SUMOylation pathway components, including SMT3 (SUMO), AOS1 (an E1 enzyme), UBC9 (an E2 enzyme), and MMS21 (an E3 ligase), in *Fusarium oxysporum* f. sp. *niveum* (*Fon*), the causative agent of watermelon Fusarium wilt, in terms of the phylogenetic relationship, gene/protein structures, and basic biological functions. The SUMOylation components FonSMT3, FonAOS1, FonUBC9, and FonMMS21 are predominantly located in the nucleus. *FonSMT3*, *FonAOS1*, *FonUBC9*, and *FonMMS21* are highly expressed in the germinating macroconidia, but their expression is downregulated gradually in infected watermelon roots with the disease progression. The disruption of *FonUBA2* and *FonSIZ1* seems to be lethal in *Fon*. The deletion mutant strains for *FonSMT3*, *FonAOS1*, *FonUBC9*, and *FonMMS21* are viable, but exhibit significant defects in vegetative growth, asexual reproduction, conidial morphology, spore germination, responses to metal ions and DNA-damaging agents, and apoptosis. The disruption of *FonSMT3*, *FonAOS1*, *FonUBC9*, and *FonMMS21* enhances sensitivity to cell wall-perturbing agents, but confers tolerance to digestion by cell wall-degrading enzymes. Furthermore, the disruption of *FonSMT3*, *FonAOS1*, and *FonUBC9* negatively regulates autophagy in *Fon*. Overall, these results demonstrate that the SUMOylation pathway plays vital roles in regulating multiple basic biological processes in *Fon*, and, thus, can serve as a potential target for developing a disease management approach to control Fusarium wilt in watermelon.

## 1. Introduction

SUMOylation is a dynamic and reversible multistep post-translational modification process that attaches small ubiquitin-related modifiers (SUMOs) to substrate proteins [1]. This process often regulates protein functions by modifying their subcellular fate, biochemical activity, or protein–protein interaction capability [1,2,3]. SUMOylation is a highly conserved process that is essential for the growth, development, and adaptation to multiple stresses in almost all eukaryotes, including fungi [1]. In contrast to other post-translational modifications, such as phosphorylation, methylation, and ubiquitination, the SUMOylation process in phytopathogenic fungi is poorly understood.

SUMOs are a family of small proteins comprising ~100–115 amino acids with molecular weights of ~11 kDa, and possess flexible amino acid extensions at their N-terminal [2,4]. Unlike human and *Arabidopsis* [5,6], most fungi have only one SUMO encoded by a single gene [1]: for example, SMT3 in *Saccharomyces cerevisiae* [7] and MoSMT3 in *Magnaporthe oryzae* [8,9]. However, some fungi, including the phytopathogenic fungus *Botrytis cinerea*, possess two or more SUMOs [1]. In *S. cerevisiae*, the disruption of *SMT3* is lethal [7]; however, SUMO is not essential for the viability of other fungi. For example, the inactivation of *SMT3* in *Schizosaccharomyces pombe*, *Candida albicans*, *Aspergillus nidulans*, *Aspergillus flavus*, *M. oryzae*, and *Beauveria bassiana* resulted in growth- and development-related defects [8,9,10,11,12,13,14].

SUMOylation of the substrate proteins involves a sequential concerted action of a cascade of enzymes including SUMO-activating enzymes (E1), SUMO-conjugating enzymes (E2), SUMO ligases (E3), and SUMO proteases [1]. In *S. cerevisiae*, the SUMOylation pathway consists of E1 heterodimer activating enzymes (AOS1/UBA2), an E2 conjugation enzyme (UBC9), and various E3 ligases (e.g., SIZ1, SIZ2, Cst9, and MMS21) [1,15,16]. In the SUMOylation process, the SUMO protease converts the SUMO precursor into its mature form by exposing the C-terminal diglycine motif [17]. Subsequently, a thioester bond is formed between a cysteine residue of E1 and a glycine residue of a mature SUMO [1]. Then, SUMO is transferred from E1 to a cysteine residue of E2 via a thioester bond [18]. E3 assists in conjugating a glycine residue of SUMO with a lysine residue of a consensus or a non-consensus motif on the substrate through an isopeptide bond [19]. The SUMO-specific proteases, such as Ulp1 and Ulp2, finally release reusable SUMO from its substrate [20,21,22].

The SUMOylation pathway has been shown to modulate a variety of cellular and biochemical processes, including chromosome segregation, DNA replication, cell cycle events, telomere position effect, septin ring, and nuclear pore dynamics [1,16]. Given facts suggest that SUMOylation is essential for distinct biological functions, such as transcriptional activity, protein localization/stability, cell cycle, programmed cell death, DNA-damage repair, and stress tolerance [1,16]. In *S. cerevisiae*, SMT3, AOS1, UBA2, UBC9, MMS21, and ULP1 have been found to be crucial for cell viability [23]. CaSMT3, CaAOS1, and CaMMS21 have been demonstrated to play regulatory roles in cell growth/morphology, cell wall integrity, and stress response in *C. albicans* [11,24]. In *A. nidulans*, SUMO has been shown to be non-essential for fungal vegetative growth, but indispensable for cellular differentiation [25]. The inactivation of *ULPA* and *ULPB* resulted in growth, conidiation, and self-sterility defects in *A. nidulans* [12,25]. The disruption of the *A. flavus* gene, *AfsumO*, caused significant defects in conidiation [13]. In *M. orayzae*, the deletion of *MoSMT3*, *MoAOS1*, *MoUBA2*, and *MoUBC9* led to significant defects in mycelial growth, conidiation, septum formation, cell cycle, conidial germination, appressorium formation, and pathogenicity, and also attenuated the tolerance to nutrient starvation, DNA damage, and other environmental stimuli [8,9,26]. Similarly, the deletion of the SUMO components led to reductions in virulence and tolerance to DNA damage agents in *Fusarium graminearum* [27]. These investigations demonstrate that the SUMOylation pathway and its components play important roles in the growth, development, stress response, and virulence of phytopathogenic fungi. The role of the SUMO pathway in stress response and virulence has been established in *F. graminearum* [27]; however, the basic biological functions, such as growth and development, of the SUMOylation pathway need to be studied in the *Fusarium* species.

Watermelon Fusarium wilt, caused by *Fusarium oxysporum* f. sp. *niveum* (*Fon*), is one of the most destructive fungal diseases. In the present study, we identified the putative SUMOylation pathway components in *Fon* and revealed their phylogenetic relationship, as well as the gene and protein structures. Further, we investigated their roles in the basic biological functions of *Fon* by generating the targeted deletion mutants. Our results showed that the SUMOylation pathway is essential for the mycelial growth, conidiation, conidial morphology, spore germination, cell wall integrity, responses to DNA damage or metal ions stresses, apoptosis, and autophagy of *Fon*, thus highlighting its significance in regulating imperative biological functions in plant pathogenic fungi.

## 2. Materials and Methods

### 2.1. Fungal Strain and Growth Conditions

*Fon* strain ZJ1 was used as the wild type (WT) strain for the genetic manipulation. The following media were used for the growth and stress sensitivity tests: minimal medium (MM; 0.5 g KCl, 2 g NaNO_3_, 1 g KHPO_4_, 0.5 g MgSO_4_·7H_2_O, 0.01 g FeSO_4_·7H_2_O, 30 g dextrose, pH 6.5, 1 L ddH_2_O), complete medium (CM; 2 g peptone, 6 g NaNO_3_, 10 g glucose, 0.52 g KCl, 0.52 g MgSO_4_·7H_2_O, 1.52 g KH_2_PO_4_, 1 g casamino acids, 1 g yeast extract, 0.01 mg vitamins, 0.01 mg trace elements, pH 6.5, 1 L ddH_2_O), potato dextrose agar (PDA) medium (200 g potato, 20 g dextrose, 20 g agar, 1 L ddH_2_O), yeast extract peptone dextrose (YEPD) medium (3 g yeast extract, 10 g peptone, 20 g dextrose, pH 7.0, 1 L ddH_2_O), and mung bean liquid (MBL) medium (15 g mung bean, 1 L ddH_2_O).

### 2.2. Bioinformatics Analyses

To identify the SUMOylation pathway components of *Fon*, BLASTp was performed using *M. oryzae* SMT3, UBC9, AOS1, and MMS21 sequences [8,9] as queries, and then searched against the *F. oxysporum* genome database FungiDB (https://fungidb.org/fungidb/app, accessed on 10 February 2019). To determine the phylogenetic relationship, the retrieved protein sequences of the SUMOylation pathway components in *Fon* were aligned with those from representative fungal species and humans using the MUSCLE program [28]. A phylogenetic tree was constructed with the maximum likelihood method through the MEGA6.0 (Molecular Evolutionary Genetics Analysis) program using a bootstrap approach [29]. For gene structure analysis, full-length genomic and coding sequences of each gene were uploaded to the Gene Structure Display Server (GSDS) website (http://gsds.cbi.pku.edu.cn/, accessed on 24 October 2022) [30], and the resultant structure of each gene was obtained. Conserved motifs and their organization were obtained using the Multiple Em for Motif Elicitation (MEME) tool (http://meme-suite.org/tools/meme, accessed on 24 October 2022) with the default parameters, except for any number of repetitions and a maximum number of motifs of 15 [31].

### 2.3. Generation of Targeted Deletion and Complementation Strains

To generate the gene replacement constructs, the 5′ and 3′ flanking sequences of the SUMOylation pathway genes were amplified from *Fon* genomic DNA and fused with the 1349 bp hygromycin gene using the double-joint PCR method. The obtained gene knockout fragments were directly transformed into WT protoplasts [32,33]. The resulting transformants were screened with 100 µg/mL hygromycin B, and were further validated by polymerase chain reaction (PCR), reverse transcriptase (RT)-quantitative PCR, and Southern blotting assays. To generate the complementation strains, vectors were constructed as previously described [34]. As the C-terminal of SUMO protein is required for the attachment of substrates, a complementation vector of *FonSMT3* with *Green Fluorescent Protein* (*GFP*) fusion at the N-terminus, under the control of the *Rp27* promoter, was generated. The *GFP* gene was amplified from pYF11 plasmid with primers SMT3-nPYF111p1-F and SMT3-nPYF11p1-R (Appendix A), while the *FonSMT3* ORF fragment was amplified from genomic DNA with primers SMT3-nPYF11p2-F and SMT3-nPYF11p2-R (Appendix A). The *GFP* and *FonSMT3* ORF fragments were fused by the double joint PCR method, and co-transferred with *Xho*I-digested pYF11-neo plasmid into yeast XK1-25 cells. To generate the complementation vectors for *FonAOS1*, *FonUBC9*, and *FonMMS21*, the coding fragments of the respective genes were amplified from genomic DNA and then co-transformed with *Xho*I-digested pYF11-neo plasmid into yeast XK1-25 cells. For each gene, the plasmid from a positive yeast colony was then transformed into *E. coli* DH5α cells. The generated complementation vectors were individually transformed into the protoplasts of the respective deletion mutants. Transformants were initially screened with 50 µg/mL neomycin, and were further verified by PCR and RT-qPCR. All primers used are listed in Appendix A.

### 2.4. Fungal Growth, Conidiation, and Stress Tolerance Assays

The *Fon* strains, including WT, deletion mutants, and complementation strains, were grown on PDA or MM supplemented with/without various stress-inducing compounds, and colony sizes were measured [33]. For the conidiation assays, the strains were incubated on MM or PDA at 26 °C for 7 d, and the conidia were collected and counted as previously described [33]. The conidiation data were then normalized by dividing the conidia number by the colony area, and presented as the conidia per cm^2^. For the spore germination assays, the *Fon* strains were grown in MBL for 2 d, and macroconidia were collected and transferred into YEPD with/without stresses. After incubation for 6 or 12 h at 26 °C, microscopic examination of at least 100 randomly selected spores per field was carried out to determine the spore germination status. Stress sensitivity was assessed by determining the growth inhibition rate, as previously described [33]. The chemicals or enzymes used and their sources are as follows: Congo red (CR; Sigma-Aldrich, St. Louis, MO, USA), Calcofluor white (CFW; Sigma-Aldrich), Camptothecin (CPT; Sigma-Aldrich), Methyl methane sulfonate (MMS; Sigma-Aldrich), 4-Nitroquinoline (4-NQ; Sigma-Aldrich), Hydroxyurea (HU; Sigma-Aldrich), Calcium chloride (CaCl_2_; Sigma-Aldrich), Sodium chloride (NaCl; Sinopharm Chemical, Shanghai, China), Iron chloride (FeCl_3_; Sinopharm Chemical), Zinc chloride (ZnCl_2_; Sinopharm Chemical), Copper chloride (CuCl_2_; Sinopharm Chemical), Driselase (Sigma-Aldrich), Lysozyme (Shanghai Ryon BioTech, Shangha, China), and Cellulase (Shanghai Ryon BioTech). All experiments were performed independently at least three times.

### 2.5. Fluorescence Microscopy

To analyze septation and subcellular localization, fresh conidia and mycelia of the *Fon* strains were washed with sterile ddH_2_O_2_ and co-stained with CFW or Hoechst 33342, respectively [9]. For apoptotic cell death analysis, the germ tubes of the *Fon* strains were treated with/without apoptosis-inducing compound farnesol (FOH; Shanghai Aladdin Bio-Chem, Shanghai, China) at 25 or 50 μM for 4 h at 26 °C. Necrotic cells and nuclei were co-stained with propidium iodide (PI; Shanghai Yeasen Biotech, Shanghai, China) and Hoechst 33342 (Shanghai Yeasen Biotech, China), respectively [35,36]. To examine autophagy, the *Fon* strains were cultivated in liquid CM for 12 h, and the collected mycelia were rinsed with sterile ddH_2_O_2_. After washing, the mycelia were transferred into nitrogen-starved MM medium (MM-N) with/without 4 mM phenylmethanesulfonyl fluoride (PMSF; Sigma-Aldrich, St. Louis, MO, USA) for 1 h. The mycelia were stained with the fluorescent dye monodansylcadaverine (MDC; Sigma-Aldrich, St. Louis, MO, USA) for 30 min in the dark, followed by ddH_2_O_2_ washing [37,38]. All aforementioned samples were observed under a Zeiss LSM 780 Meta confocal microscope (Gottingen, Niedersachsen, Germany) with the appropriate excitation and emission filters for corresponding dyes and signals.

### 2.6. Transmission Electron Microscopy (TEM)

TEM observations for cell wall thickness and autophagy were conducted as previously described [39]. Fresh mycelia grown in nitrogen-starved or nitrogen-rich conditions were collected and fixed with 2.5% (*v*/*v*) glutaraldehyde. The samples were dehydrated in a graded series of ethanol and then immersed in epoxy resin. The specimens were sectioned into ultrathin sections using an EM UC7 ultramicrotome (Leica Microsystems, Vienna, Austria), and stained with uranyl acetate or lead citrate. The stained ultrathin sections were subjected to TEM (H-7650; Hitachi, Japan).

### 2.7. RT-qPCR Analyses

Total RNA was extracted by immersing pulverized samples in RNA isolator reagent (Vazyme Biotech, Nanjing, China) and treated with RNase-free DNase according to the manufacturer’s instructions. Subsequently, cDNA was synthesized from 1 µg total RNA using HiScript QRT SuperMix (Vazyme Biotech, Nanjing, China). The reaction mixture of qPCR was prepared using AceQ qPCR SYBR Green Master Mix (Vazyme Biotech, Nanjing, China), and qPCR was carried out with three technical replicates using a CFX96 real-time PCR system (BioRad, Hercules, CA, USA). *FonActin* was used as an internal control to normalize the qPCR data for the purpose of comparing the relative transcript abundance of the target genes, and relative expression of the genes was calculated using the 2^−∆∆CT^ method [40]. The gene-specific RT-qPCR primers are listed in Appendix A.

### 2.8. Western Blotting Assays

Fresh mycelia of the Δ*Fonsmt3*::*GFP-FonSMT3* strain were quickly pulverized in liquid nitrogen and resuspended in 1 mL lysis buffer (50 mM Tris-HCl, 100 mM NaCl, 5 mM EDTA, 1% Triton X-100, 10 μL PMSF, pH7.5) containing 40 μL 25× protease inhibitor cocktail. The suspension was then centrifuged at 13000 rpm and the supernatant was collected. Western blotting hybridization was performed as previously described [32]. The immunoblot experiments were conducted using an anti-GFP antibody (Abcam, Cambridge, MA, USA). GAPDH, immunoblotted with an anti-GAPDH antibody (Huaan Biotechnology, Hangzhou, China), was used as the reference.

### 2.9. Statistical Analysis

All experiments were independently performed three times. Data were normalized when necessary and analyzed using one-way analysis of variance (ANOVA). The significance of the various data was determined by the Fisher’s least significance difference test at a 95% confidence level.

## 3. Results

### 3.1. Identification of the SUMOylation Pathway Components in Fon

To investigate the structural characteristics of the SUMOylation pathway in *Fon*, we first identified the SUMOylation pathway components in *F. oxysporum*, and found all of the orthologs of *M. oryzae* SUMOylation pathway proteins in *F. oxysporum* (Appendix A). *F. oxypsorum* SMT3 and UBC9 exhibited strong sequence similarities (>75%) to those in *M. oryzae* (Appendix A). The other *F. oxysporum* SUMOylation components, including AOS1, UBA2, SIZ1, and MMS21, also showed significant sequence similarities of 55%, 61%, 51%, and 35 %, respectively, to those in *M. oryzae* (Appendix A). The *F. oxysporum* SMT3 possesses a SUMO domain (PF11976; Appendix A); AOS1 harbors a Thif domain (PF00899; Appendix A); UBA2 contains three different conserved domains, namely Thif (PF00899), UAE (PF14732), and UBA (PF10585); UBC9 carries a ubiquitin-conjugating domain (PF00179; Appendix A); SIZ1 comprises various zinc finger domains (PF02891 and PF14634); and MMS21 bears a SP-RING-type domain (PF11789) (Appendix A).

In the phylogenetic analysis, the *F. oxysporum* SUMOylation pathway components were shown to share close relationships with the orthologs in the filamentous fungi such as *Fusarium proliferatum* (Fp), *M. oryzae* (Mo), *Colletotrichum graminicola* (Cg), *B. cinerea* (Bc), *A. flavus* (Af), *A. nidulans* (An), and *Blumeria graminis* (Bg) (Figure 1A). Further, the *F. oxysporum* SUMOylation components showed certain degrees of similarities to the orthologs in single-celled fungi (e.g., *S. cerevisiae* [Sc], *S. pombe* [Sp], and *C. albicans* [Ca]), oomycete *Phytophthora infestans* (Pi), and *Homo sapiens* (Hs) (Figure 1A). The gene structure analysis revealed similar exon/intron numbers and organization in the majority of the closest members of the *SMT3*, *AOS1*, *UBC9*, and *MMS21* groups, with the exception of a few members (Figure 1B, Appendix A). For instance, most of the filamentous fungi had two exons and one intron in the *SMT3*, *AOS1*, and *MMS21* groups, and three introns and four exons in the *UBC9* group (Figure 1B, Appendix A). However, members of Sc, Sp, Ca, Pi, and Hs, in all groups, showed a smaller to greater extent of heterogeneity, either in the gene lengths or in the exon/intron number and organization (Figure 1B, Appendix A). These features of the gene structure offer reliable evidence to support the phylogenetic groupings of *F. oxysporum:* SMT3, AOS1, UBC9, and MMS21 (Figure 1A).

MEME analysis identified fifteen distinct motifs in *F. oxysporum* SUMOlyation pathway components, which were labeled consecutively as motifs 1–15 (Figure 1C and Appendix A). Seven motifs, including 1, 2, 3, 6, 7, 8, and 11, were recognized as UQ_con, Ubiquitin, ThiF, KYD|A, ThiF, UQ_con, and zf-RING_UBOX, respectively (Figure 1C), while the others, including motifs 4, 5, 9, 10, 12, 13, 14, and 15, were unknown. Based on the phylogenetic grouping, most of the members of the SUMOylation component groups shared common motifs, with no/minor variations in alignment or position among the closely related members. For instance, most of the members of the SMT3 group contained two motifs (2 and 10), except the orthologs in Sp, Pi, and Hs, which possessed only motif 2 (Figure 1C). Contrasting to Hs, Pi, Ca, Sc, and Sp, the majority of the members of the AOS1 group shared eight similar motifs, including three characterized motifs: 3, 6, and 7 (Figure 1C). Importantly, motifs 3 and 7 were common in all AOS1 group members, except CaAOS1, which contained only motif 3 (Figure 1C). In contrast to Sc and Sp, all members of the UBC9 group possessed four common motifs, including 14, 9, 1, and 8 (Figure 1C). In the MMS21 group, most of the members shared two common motifs, labeled as 11 and 12; however, Sc, Sp, and Hs had only motif 11 (Figure 1C). Among all 15 motifs, motif 12 was present in most members of the AOS1 and MMS21 groups (Figure 1C and Appendix A). Overall, these results indicate that the *F. oxysporum* SUMOylation pathway components share similar gene structures and motif organization with the closest orthologs, and, thus, can perform similar functions.

### 3.2. Expression Profiling of the SUMOylation Pathway Genes in Fon

To explore the role of the SUMOylation pathway in the development and infection process in *Fon*, we first examined the expression levels of the SUMOylation pathway genes in *Fon* at different development stages, including mycelia, conidia, and germinating macroconidia, and during root infection. *FonSMT3*, *FonAOS1*, and *FonMMS21* showed comparable expressions in the mycelia and macroconidia of *Fon*, while the *FonUBC9* showed significantly higher expression in macroconidia than in the mycelia (Figure 2A–D). The expression levels of *FonSMT3*, *FonAOS1*, *FonUBC9*, and *FonMMS21* were significantly elevated in the germinating macroconidia at 12 h post-germination compared to the mycelia, although *FonSMT3* and *FonMMS21* had comparable expressions in the germinating macroconidia at 6 h post-germination to that in the mycelia (Figure 2A–D). Importantly, the expression levels of *FonSMT3*, *FonAOS1*, *FonUBC9*, and *FonMMS21* in infected watermelon roots at 3 d post-inoculation were significantly higher than those in the mycelia or macroconidia, and gradually decreased as the disease progressed (Figure 2A–D). These results suggest that *FonSMT3*, *FonAOS1*, *FonUBC9*, and *FonMMS21* are differentially expressed in macroconidia, germinating macroconidia, and during the infection process, implying a potential role of the SUMOylation pathway in regulating *Fon* pathogenicity towards watermelon plants.

### 3.3. Generation and Characterization of the Deletion Mutants and Complementation Strains

To genetically evaluate the biological functions of the SUMOylation pathway components in *Fon*, we obtained the deletion mutants for *FonSMT3*, *FonAOS1*, *FonUBC9*, and *FonMMS21* through the homologous recombination strategy, and named them Δ*Fonsmt3*, Δ*Fonaos1*, Δ*Fonubc9*, and Δ*Fonmms21*, respectively. However, despite several attempts, we were unable to obtain the deletion mutants for *FonUBA2* and *FonSIZ1* (Appendix A), suggesting that these two genes might be essential for *Fon* survival and viability. The deletion mutant strains Δ*Fonsmt3*, Δ*Fonaos1*, Δ*Fonubc9*, and Δ*Fonmms21* were confirmed through Southern blotting and PCR assays by detecting the inserted *HPH* cassette containing knockout constructs. The complementation strains Δ*Fonsmt3*-C, Δ*Fonaos1*-C, Δ*Fonubc9*-C, and Δ*Fonmms21*-C were obtained for each of the corresponding deletion mutants. RT-qPCR analysis revealed that the transcripts of *FonSMT3*, *FonAOS1*, *FonUBC9*, and *FonMMS21* were undetectable in the corresponding deletion mutant strains, but were comparable or higher in the respective complementation strains than WT.

### 3.4. Subcellular Localzation of the SUMOylation Pathway Components in Fon

The majority of the SUMOylation pathway components are localized in the nucleus, and occasionally in the cytoplasm [9,41,42,43]. To investigate the functional site and accumulation of SUMOylation pathway components in *Fon*, we studied the subcellular localization of GFP-tagged FonSMT3, FonAOS1, FonUBC9, and FonMMS21. FonSMT3 and FonMMS21 were largely detected in the nucleus of mycelia and macroconidia (Figure 2E,F). Conversely, FonUBC9 and FonAOS1 were mostly found in the nucleus, but at a low level, in the cytoplasm of mycelia and macroconidia (Figure 2E,F). These results indicate that the SUMOylation pathway components, including FonSMT3, FonAOS1, FonUBC9, and FonMMS21, mainly localize and accumulate in the nuclei of the mycelia and macroconidia of *Fon* for their functions.

### 3.5. The SUMOylation Pathway Regulates the Growth and Development in Fon

To examine the role of the SUMOylation pathway in the growth and development of *Fon*, we performed comparative phenotypic analyses of the WT, deletion mutants, and complementation strains. When grown on PDA or MM, the mycelial growth of the deletion mutants was significantly reduced, with the most obvious change in Δ*Fonmms21*, leading to a decrease of 56% and 60% on PDA and MM, respectively, compared to WT (Figure 3A–C). In contrast to WT, which had normal branches at the hyphal tip, the deletion mutants showed dispersed hypha with fewer or no branches at the hyphal tips (Figure 3D). Microscopic observations showed that the deletion mutants produced significantly fewer macroconidia than WT when cultured on PDA or MM (Figure 4A,B). The macroconidia of the deletion mutants lacked the normal morphology (Figure 4C), and were shorter in length than WT (Figure 4D). The CFW staining assay revealed that 71%, 83%, 68%, and 39% of the macroconidia produced by the Δ*Fonsmt3*, Δ*Fonaos1*, Δ*Fonubc9*, and Δ*Fonmms21* strains, respectively, had no septum, which was significantly higher than WT (which had 14% conidia without septum) (Figure 4C,E). By contrast, 2%, 4%, 3%, and 26% of the macroconidia from the Δ*Fonsmt3*, Δ*Fonaos1*, Δ*Fonubc9*, and Δ*Fonmms21* strains had three septa, respectively, which was remarkably lower than WT (which produced 46% macroconidia with three septa) (Figure 4C,E). Spore germination rates of the deletion mutants were significantly lower than those of WT at 6 and 12 h post-germination (Figure 4F). The deletion mutants seemed to have significantly shorter hyphal cells than WT (Figure 4G,H). All of these defects in the mycelial growth, conidiation, conidial morphology, and spore germination in the deletion mutants were rescued completely or partially in the corresponding complementation strains Δ*Fonsmt3*-C, Δ*Fonaos1*-C, Δ*Fonubc9*-C, and Δ*Fonmms21*-C (Figure 3A–D and Figure 4A–F). Notably, the partial growth or phenotypic recovery of the complementation strains might be due to the non-physiological expression of the genes that was driven by the *Rp27* promoter rather than their native promoters. These results suggest that the SUMOylation pathway is essential for the vegetative growth, conidiation, spore germination, and conidial morphology of *Fon*.

### 3.6. The SUMOylation Pathway Maintains Cell Wall Integrity in Fon

The SUMOylation pathway is critical for stress responses in fungi, including *M. oryzae* [8,44]. To test whether the SUMOylation pathway has a role in the maintenance of *Fon* cell wall integrity, we analyzed the effect of three cell wall-perturbing agents, including CR, CFW, and NaCl [33], on the mycelial growth of the WT, deletion mutants, and complementation strains. When grown on PDA supplemented with 0.2 mg/mL CR, 0.1 mg/mL CFW or 0.5 M NaCl, mycelial growth of the deletion mutants was significantly inhibited compared to WT (Figure 5A–D). Surprisingly, the deletion mutants exhibited an enhanced resistance to cell wall-degrading enzymes (CWDEs) compared with WT. After 30, 60, or 90 min of CWDE digestion, the released protoplasts from the hypha of the deletion mutants were significantly less compared to WT (Figure 5E–H). Furthermore, TEM examination confirmed that the enhanced resistance to CWDE in the deletion mutants was due to a change in the cell wall structure, leading to an abnormal increase in the cell wall thickness (Figure 5I,J). The defects in the cell wall integrity and structure upon exposure to cell wall-perturbing agents and CWDE digestion were recovered in the complementation strains, showing phenotypes comparable to WT (Figure 5A–J). Furthermore, we examined the protein level of GFP-FonSMT3 in the Δ*Fonsmt3*-C strain grown in potato dextrose broth (PDB) supplemented without or with different cell wall-perturbing agents, including CR, CFW, and NaCl. Western blotting results showed relatively higher levels of GFP-FonSMT3 in Δ*Fonsmt3*-C strain under stressed conditions (Figure 5K), suggesting that cell wall-perturbing agents can activate the expression and accumulation of FonSMT3 in *Fon*. These results indicate that the SUMOylation pathway is involved in the maintenance of cell wall integrity in *Fon*.

### 3.7. The SUMOylation Pathway Orchestrates DNA Damage Responses in Fon

To study the role of the SUMOylation pathway in DNA damage responses in *Fon*, we analyzed the effect of four DNA-damaging agents, namely CPT (0.45 µg/mL), MMS (0.13 mg/mL), HU (1 mg/mL), and 4-NQ (10 µg/mL), on the mycelial growth of the WT, deletion mutants, and complementation strains. Phenotypic tests showed that the deletion mutants exhibited increased sensitivity to DNA-damaging agents as compared to WT (Figure 6A–E). Specifically, Δ*Fonsmt3* and Δ*Fonaos1* were highly sensitive to CPT (Figure 6A,B), while Δ*Fonmms21* was extremely sensitive to MMS (Figure 6A,C). The abnormality in the sensitivity responses of the deletion mutants to DNA-damaging agents was recovered in the corresponding complementation strains (Figure 6A–E). Furthermore, we examined the effect of these DNA-damaging agents on the accumulation of GFP-FonSMT3 in Δ*Fonsmt3*-C. Immunoblot analysis using the anti-GFP antibody showed an increased accumulation of GFP-FonSMT3 in Δ*Fonsmt3*-C grown in PDB supplemented with the aforementioned DNA-damaging agents compared to the levels in the untreated control (Figure 6F), indicating that the DNA-damaging agents can trigger the expression and accumulation of GFP-FonSMT3 in *Fon*. These results suggest that the SUMOylation pathway is crucial for *Fon*‘s response to DNA damage. 

### 3.8. The SUMOylation Pathway Affects Metal ion Responses in Fon

To determine whether the SUMOylation pathway modulates metal ion response in *Fon*, we analyzed the growth phenotype and inhibition rate of the WT, deletion mutants, and complement strains in the presence of the effects of four metal cations, Fe^3+^ (1 mM), Zn^2+^ (0.5 mM), Ca^2+^ (0.2 M), and Cu^2+^ (0.5 mM). The mycelial growth of WT was inhibited by the tested metal cations (Figure 7A–E), implying that the levels of metal ions used in our study were above physiological limits. Notably, the mycelial growth of the deletion mutants was remarkably inhibited by Fe^3+^, Ca^2+^, and Cu^2+^ compared to WT (Figure 7A,B,D,E). Precisely, Zn^2+^ suppressed the mycelial growth of Δ*Fonsmt3* and Δ*Fonmms21*, but not that of Δ*Fonaos1* and Δ*Fonubc9* (Figure 7A,C). Overall, the deletion mutants were found to be more sensitive to Ca^2+^ and Cu^2+^, but showed less or no change in sensitivity to Fe^3+^ and Zn^2+^, as compared to WT (Figure 7A–E). Among the deletion mutants, Δ*Fonaos1* had the highest level of growth inhibition under 0.2 M Ca^2+^ and 0.5 mM Cu^2+^ stress (Figure 7A,D,E). The sensitivity of the elevated metal ions in the deletion mutants was restored in the corresponding complementation strains (Figure 7A–E). These results suggest that the SUMOylation pathway is important for the activation of metal ion detoxification mechanisms in *Fon,* with a shared and distinct sensitivity to different metal ions.

### 3.9. The SUMOylation Pathway Negativelty Regulates Apoptosis in Fon

The SUMOylation pathway is associated with various cell death processes, including apoptosis [45,46,47,48]. To explore the effect of the SUMOylation pathway on *Fon* apoptosis, we compared the mycelial growth of the WT, deletion mutants, and complement strains in response to an apoptosis-inducing compound, FOH, at 25 or 50 µM. When grown on PDA supplemented with 25 or 50 μM FOH, the mycelial growth of the deletion mutants was significantly inhibited compared to WT and the corresponding complementation strains (Figure 8A–C). More precisely, the inhibition of mycelial growth by FOH was very obvious for the Δ*Fonmms21* strain (Figure 8A–C). FOH, at 25 μM, significantly inhibited the germination of macroconidia in the deletion mutants compared to WT (Figure 8D). The growth and developmental defects of the deletion mutants under FOH stress were restored in the corresponding complementation strains (Figure 8A–D). Furthermore, Western blotting assays revealed an increased accumulation of GFP-FonSMT3 in Δ*Fonsmt3*-C grown in PDB supplemented with 25 μM FOH compared to the level without FOH (Figure 8E), indicating that the apoptosis-inducing agent can trigger the expression and accumulation of GFP-FonSMT3 in *Fon*. To further confirm the role of the SUMOylation pathway in apoptotic cell death in *Fon*, the macroconidia of the WT, deletion mutants, and complementation strains were allowed to germinate in the presence of FOH (25 μM or 50 μM), and the resulting germ tubes were stained with Hoechst 33342 and PI. At the 25 μM FOH level, the germ tubes of the deletion mutants showed obvious apoptosis- and necrosis-like cell features, such as chromatin condensation and cell death, whereas the germ tubes of WT and complementation strains did not show any obvious change (Figure 9). FOH, at 50 μM, was extremely toxic to all tested *Fon* strains, resulting in chromatin condensation, marginalization, and severe necrosis; however, apoptotic cell death was much evident in the germ tubes of the deletion mutants, as judged by the more prominent PI staining (Figure 9). Collectively, these results suggest that the SUMOylation pathway controls apoptotic-like cell death, and thus maintains the cellular stability of *Fon*.

### 3.10. The SUMOylatin Pathway Mediates Autophagy in Fon

Previous studies have shown that the SUMOylation pathway is involved in autophagy initiation [45,49,50]. Therefore, we determined whether the SUMOylation pathway is involved in autophagy in *Fon,* and found that the MDC-stained mycelia of all tested strains did not show any obvious autophagic compartments under nitrogen-rich conditions. However, Δ*Fonsmt3*, Δ*Fonaos1*, and Δ*Fonubc9* showed less induction of autophagic compartments under nitrogen-starved conditions in the presence or absence of PMSF, compared to WT (Figure 10A). By contrast, Δ*Fonmms21* showed no significant difference in autophagy induction compared to WT when grown in nitrogen-starved conditions, either in the presence or absence of PMSF (Figure 10A). Further, ultrastructural observations of autophagic features, e.g., autophagosome and autophagic body, by TEM showed no observable change in the hyphal cells of WT or the deletion mutants under nitrogen-rich conditions (Figure 10B). However, in comparison to WT, the hyphal cells of Δ*Fonsmt3*, Δ*Fonaos1*, and Δ*Fonubc9* displayed fewer multilamellar cupped membrane or membrane-surrounded autophagic structures under nitrogen-starved conditions in the presence or absence of PMSF, whereas Δ*Fonmms21* showed comparable results to WT under both conditions (Figure 10B). Together, these observations indicate that FonSMT3, FonAOS1, and FonUBC9, to either a greater or lesser extent, are involved in autophagy in *Fon*.

## 4. Discussion

SUMOylation is a protein modification process that regulates various biological processes, such as stress tolerance, transcriptional regulation, protein localization/stability, DNA-damage repair, and cell death [1,16,21,45]. The SUMOylation pathway has been demonstrated to play critical roles in phytopathogenic fungi, such as *M. oryzae* [8,9]; however, its biological role in *F. oxysporum*, a destructive Fusarium wilt pathogen infecting a wide range of economically important crop plants, remains unclear. In this study, we describe the biological functions of the SUMOylation pathway components in regulating the vegetative growth, asexual reproduction, conidial morphology, cell wall integrity, stress responses, and programmed cell death events of *Fon*.

Previous studies have shown that SUMOylation machinery was largely localized in the nucleus, and less likely in the cytoplasm [9,41,42,43]. For instance, SMT3 in *A. flavus* and *M. oryaze* was reported to be predominantly located in the nucleus, and to some extent in the cytoplasm [8,9,13]. In *M. oryaze*, MoAOS1 and MoUBA2 were found to be localized in the nucleus, but UBC9 has been shown to be present in the nucleus and cytoplasm [9]. In plants, nuclear localization was witnessed for SIZ1 [51,52] and MMS21 [53,54]. In yeast, ULP1 was detected in the nuclear envelope, while ULP2 was present in the nucleoplasm [25]. Consistent with previous studies, we observed similar subcellular localization of the SUMOylation pathway components, FonSMT3, FonAOS1, FonUBC9, and FonMMS21 (Figure 2E,F). Thus, it is likely that the SUMOylation machinery is exclusively located in the nucleus for its function.

Similarly to previous studies [8,10,11,13,14,25], most of the SUMOylation pathway genes in *Fon*, including *FonSMT3*, *FonAOS1*, *FonUBC9*, and *FonMMS21*, were found to be non-essential for viability, but their disruption caused significant defects in mycelial growth, conidiation, conidial morphology, and spore germination (Figure 3 and Figure 4). However, the deletion of *FonSIZ1* and *FonUBA2* might be lethal, which is consistent with a previous study demonstrating the essential roles of the orthologs of these genes in maintaining cell viability in *S. cerevisiae* [55]. The disruption of SUMOylation pathway machinery may contribute to the pleiotropic characteristics of the deletion mutants, because they are involved in chromatin remodeling, transcriptional regulation, RNA synthesis, and ribosome biogenesis [56]. In this regard, it is likely that all of the SUMOylation pathway components are essential for SUMOylation as well as the regulation of different biological or biochemical functions in *Fon*. In the present study, the deletion mutants Δ*Fonsmt3*, Δ*Fonaos1*, Δ*Fonubc9*, and Δ*Fonmms21* exhibited significant defects in vegetative growth (Figure 3), which is consistent with previous studies describing the defects in mycelial growth upon deletion of the SUMOylation pathway genes in *C. albicans*, *A. nidulans*, and *M. oryzae* [8,9,11,12,14]. In our investigation, the expression levels of *FonSMT3*, *FonAOS1*, *FonUBC9*, and *FonMMS21* were significantly upregulated in the germinating macroconidia (Figure 2A–D); however, the deletion mutants were defective in conidiation and conidial germination (Figure 4F). These findings coincide with the results of previous studies on *M. oryzae*, *A. nidulans*, and *A. flavus* [9,12,13]. These data reveal that the SUMOylation pathway components are critical for the growth, conidiation, and spore germination of filamentous phytopathogenic fungi.

The structural integrity of the cell wall is essential for the growth, development, and host infection of phytopathogenic fungi, such as *Fon* [57,58]. Disruption of *SMT3* in *C. albicans* and *B. bassiana* resulted in increased sensitivity to cell wall-perturbing agents [11,14,59]. In this study, the deletion mutants displayed an enhanced resistance to CWDE digestion, probably due to the abnormal thickness of the cell wall in the deletion mutants (Figure 5). Surprisingly, the thickened cell wall did not make the deletion mutants more resistant to cell wall-perturbing agents (Figure 5). Similar observations were also reported in *C. glabrata*, where the deletion of *UlP2* attenuated the tolerance to cell wall-perturbing agents, but boosted the resistance to CWDEs [59]. These findings support the notion that the SUMOylation pathway is essential for the formation and function of the cell wall in *Fon*.

The SUMOylation pathway has been shown to be essential for DNA-damage repair and metal ion response [1,16]. The high sensitivity of the deletion mutants to the DNA-damaging agents suggests that the SUMOylation pathway may be crucial for the response to DNA damage and its subsequent repairing in *Fon* (Figure 6). This is in line with previous observations, which have noted that the deletion of the SUMOylation pathway components affected the response to DNA-damaging agents in *M. orayze*, *F. graminearum*, *A. nidulans*, and *A. flavus* [9,13,25,27]. In another study, the disruption of MMS21 altered the expressions of several DNA damage repair-associated genes in maize [60]. The crosstalk between SUMOylation and DNA damage repair in regulating genome stability and virulence in filamentous fungi has been revealed [61,62]; however, in-depth studies are required to elucidate the underlying mechanisms by which SUMO machinery regulates DNA damage response in phytopathogenic fungi. It is suggested that a functional overlap between the SUMOylation machinery and the DNA repair pathway prevents DNA-damaging challenges. Additionally, the deletion mutants showed attenuated tolerance to various metal ions (Figure 7), which is similar to the results of the *SMT3* deletion mutant in *B. bassiana* [14,35], implying that the SUMOylation machinery is involved in responses to metal ions in phytopathogenic fungi.

The SUMOylation pathway has been demonstrated to play a crucial role in a variety of programmed cell death processes, such as apoptosis and autophagy [45]. The involvement and the subsequent role of the SUMOylation pathway in autophagic and apoptotic-like cell death in plant pathogenic fungi is poorly understood. However, previous studies have demonstrated the crucial role of the SUMOylation pathway in regulating apoptosis and autophagy processes; for instance, the disruption of *SMT3*, *UBC9*, or *MMS21* promoted apoptosis in *Drosophila* and mammal cells [46,47,48]. FOH, an apoptosis-inducing compound, has been found to induce apoptotic-like cell death in fungi, including *Metarhizium robertsii* and *Hirsutella minnesotensis* [35,36]. In this study, we found that FOH at 25 μM induced an initial phase of cellular apoptosis as well as necrosis-like features in the deletion mutants (Figure 9). Furthermore, mycelial growth and conidial germination of the deletion mutants were also reduced in the presence of FOH (Figure 8). Therefore, it is likely that the disruption of the SUMOylation pathway dysregulates the FOH-induced apoptosis in *Fon*. In lung adenocarcinoma cells, the SUMOylation of eukaryotic elongation factor 2 (eEF2) has been found to be crucial for protein stability and anti-apoptotic action [63]. Further investigations are required to unravel the constitutive role of the SUMOylation machinery and eEF2 or other eukaryotic elongation factors in regulating apoptosis in *Fon*. On the other hand, overexpression of *SUMO1* or *UBC9* increased autophagic activation, while the depletion of *SUMO1* reduced the ATG12–ATG5 complex in neuroglioma H4 cells [49,64]. In *M. oryzae*, two important autophagy-related proteins, Snx41 and Atg24, were found to be putative SUMO-substrates [8], suggesting a connection between the SUMOylation pathway and autophagy in filamentous fungi. In this study, we observed that nitrogen starvation induced autophagy in *Fon*, as previously shown in *F. oxysporum* [37]. The Δ*Fonsmt3*, Δ*Fonaos1*, and Δ*Fonubc9* mutants displayed reductions in the formation of autophagosome or autophagic flux under nitrogen-starved conditions (Figure 10). Surprisingly, disruption of *FonMMS21* did not affect the formation of autophagosome or autophagic flux (Figure 10). Thus, it is likely that FonSMT3, FonAOS1, and FonUBC9 positively regulate the autophagy process. The unaltered autophagy in the Δ*Fonmms21* mutant might result from the functional compensation by another E3 ligase, FonSIZ1 (Appendix A). Further investigation is required to understand the functions and molecular mechanisms of the SUMOylation pathway components in regulating the programmed cell death processes of *Fon*.

## 5. Conclusions

In the present study, we characterized the biological functions of the SUMOylation machinery in *Fon*, and found that deletion of the SUMOylation pathway genes *FonSMT3*, *FonAOS1*, *FonUBC9*, and *FonMMS21* significantly affected multiple basic biological processes, such as vegetative growth, asexual reproduction, conidial morphology, spore germination, cell wall integrity, programmed cell death events (e.g., apoptosis and autophagy), and stress responses to DNA-damaging agents and metal ions in *Fon*. The expression levels of the SUMO components were significantly upregulated during initial infection, and gradually decreased in the infected watermelon roots at later stages of disease, thus suggesting their potential roles in *Fon* virulence. This study not only provides experimental evidence supporting the biological functions of the SUMOylation pathway components in *F. oxysporum*, but also offers an opportunity to investigate the molecular mechanisms of the SUMOylation pathway in regulating growth, development, and virulence of phytopathogenic fungi by further pinpointing the putative SUMOylation substrate proteins.

## Figures and Tables

**Figure 1 jof-09-00094-f001:**
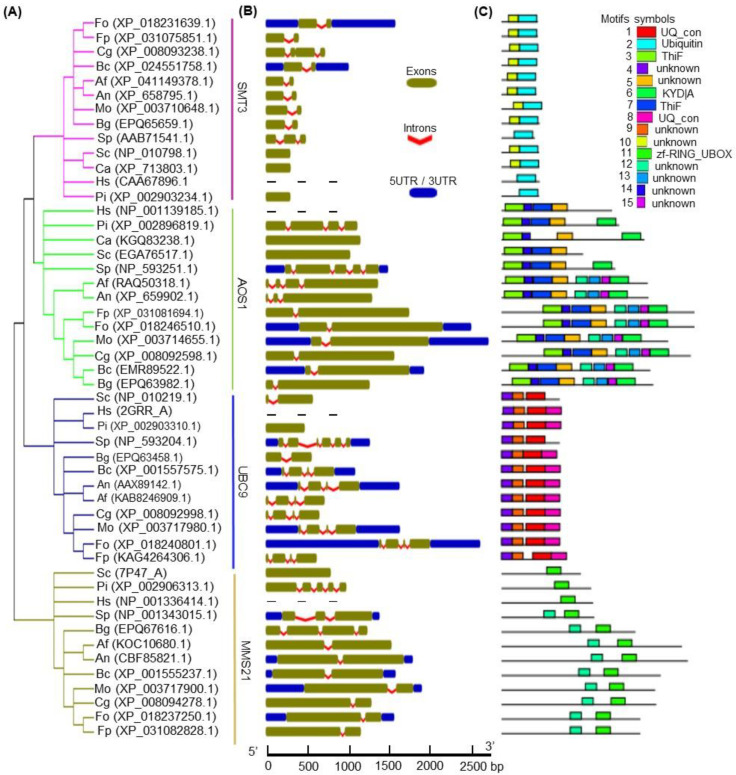
Phylogenetic relationship, gene structure, and motif organization of FonSMT3, FonAOS1, FonUBC9, and FonMMS21, as well as their orthologs from other organisms. (**A**) Phylogenetic tree. The protein sequences were aligned with MUSCLE and a maximum likelihood tree was generated using the MEGA6 software. Fo, *Fusarium oxysporum*; Fp, *Fusarium proliferatum*; Mo, *Magnaporthe oryzae*; Cg, *Colletotrichum graminicola*; Bc, *Botrytis cinerea*; Af, *Aspergillus flavus*; An, *Aspergillus nidulans*; Bg, *Blumeria graminis*; Sc, *Saccharomyces cerevisiae;* Sp, *Schizosaccharomyces pombe*; Ca, *Candida albicans*; Pi, *Phytophthora infestans*; and Hs, *Homo sapiens*. (**B**) Gene structure. The exons, introns, and upstream or downstream regions are shown by olive boxes, red spiral lines, and blue boxes, respectively. (**C**) Motifs and their organization in SMT3, AOS1, UBC9, and MMS21 proteins. Conserved motifs were identified using the MEME tool. The non-conserved sequences are shown as black lines, whereas the conserved motifs (1–15) are shown as multi-color boxes. The consensus for these conserved motifs is listed in Appendix A.

**Figure 2 jof-09-00094-f002:**
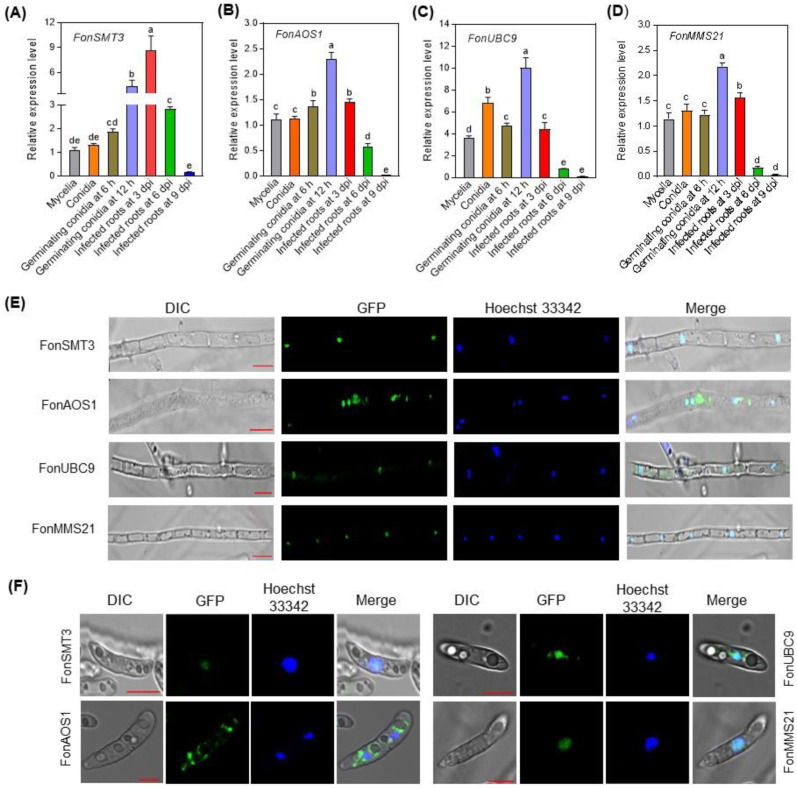
Expression profiling of *FonSMT3*, *FonAOS1*, *FonUBC9*, and *FonMMS21,* and subcellular localization of GFP-tagged FonSMT3, FonAOS1, FonUBC9, and FonMMS21 in *Fusarium oxysporum* f. sp. *niveum* (*Fon*). (**A**–**D**) Expression profiling of *FonSMT3* (**A**), *FonAOS1* (**B**), *FonUBC9* (**C**), and *FonMMS21* (**D**) at different developmental stages of *Fon* and in infected watermelon roots. (**E**,**F**) Localization of GFP-FonSMT3, FonAOS1-GFP, FonUBC9-GFP, and FonMMS21-GFP in mycelia (**E**) and macroconidia (**F**). Nuclei were stained with Hoechst 33342. Bars: 10 µm or 5 µm for mycelia (**E**) or conidia (**F**), respectively. The experiments were independently performed three times with similar results. The data shown in (**A**–**D**) are the means ± SD from three independent experiments, and the different letters above the columns denote significant differences at the *p* < 0.05 level. DIC, differential interference contrast; GFP, green fluorescent protein; dpi, d post-inoculation.

**Figure 3 jof-09-00094-f003:**
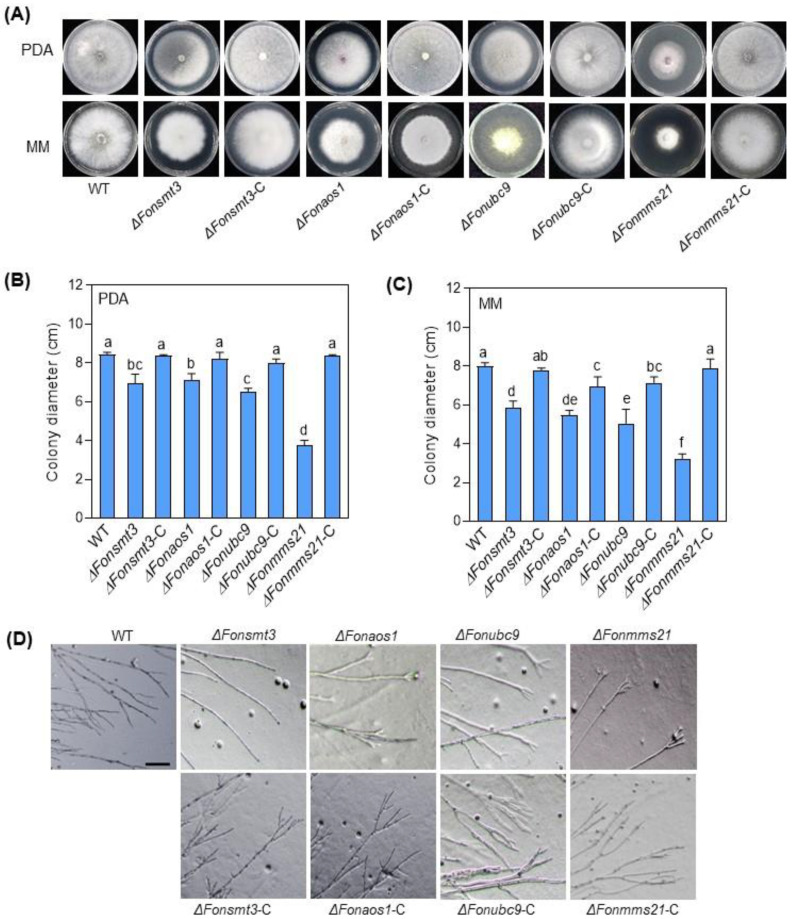
The SUMOylation pathway is essential for mycelial growth in *Fusarium oxysporum* f. sp. *niveum*. (**A**–**C**) Growth phenotype (**A**) and colony diameter of the wild type (WT), deletion mutants, and complementation strains on potato dextrose agar (PDA) (**B**) and minimal medium (MM) (**C**). Photos were taken and colony diameters were measured at 7 d post-incubation. (**D**) Hyphal branching phenotype of WT, deletion mutants, and complementation strains grown on PDA. Bar: 100 µm. The experiments were independently performed three times with similar results. The data shown in (**B**,**C**) are the means ± SD from three independent experiments, and the different letters above the columns denote significant differences at the *p* < 0.05 level.

**Figure 4 jof-09-00094-f004:**
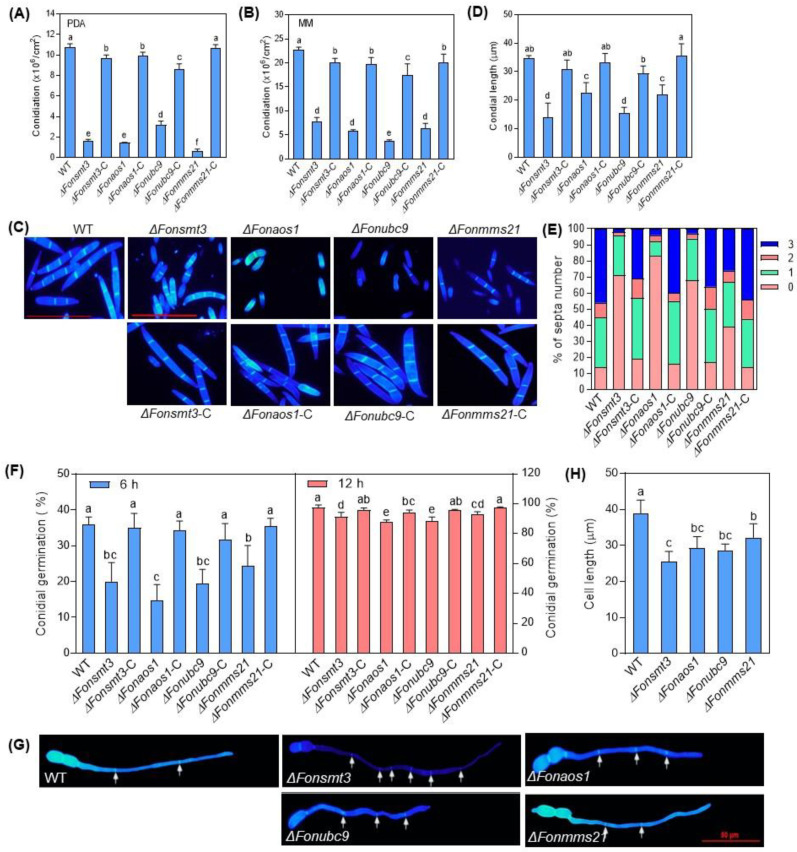
The SUMOylation pathway is essential for the conidiation, conidial morphology, and spore germination of *Fusarium oxysporum* f. sp. *niveum*. (**A**,**B**) Amounts of macroconidia produced by the wild type (WT), deletion mutants, and complementation strains grown on potato dextrose agar (PDA) (**A**) or minimal medium (MM) (**B**) plates at 7 d post-incubation. (**C**–**E**) Morphology (**C**), length (**D**), and septa number (**D**) of macroconidia. Macroconidia were stained with calcofluor white (CFW); conidial morphology and septa number were observed by differential interference contrast (DIC). Bar, 50 µm. (**F**) Spore germination. Macroconidia of WT, deletion mutants, and corresponding complementation strains were suspended in yeast extract peptone dextrose (YEPD) for 6 h (*left*) or 12 h (*right*), and spore germination was examined by the growth of germ tubes for at least 100 conidia. (**G**) Growth phenotype of germ tubes from macroconidia of WT and deletion mutants. Macroconidia were incubated in YEPD for 12 h, and the CFW-stained germ tubes were observed using DIC. Bar: 50 µm. White arrows indicate septa. (**H**) Length of the cells in germ tubes from macroconidia of WT and deletion mutants. Cell length was measured using ImageJ software. The experiments were independently performed three times with similar results. The data shown in (**A**,**B**,**D**–**F** and **H**) are the means ± SD from three independent experiments, and the different letters above the columns denote significant differences at the *p* < 0.05 level.

**Figure 5 jof-09-00094-f005:**
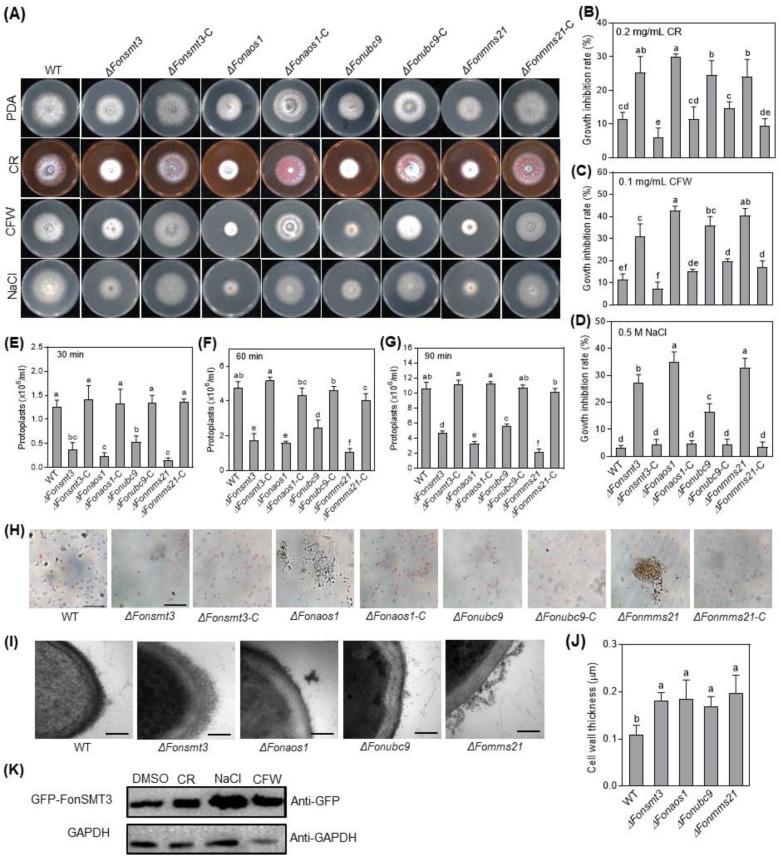
The SUMOylation pathway is involved in maintaining the cell wall integrity of *Fusarium oxysporum* f. sp. *niveum*. (**A**–**D**) Growth phenotype (**A**) and inhibition rate of mycelial growth of the wild type (WT), deletion mutants, and complementation strains when exposed to congo red (CR) (**B**), calcofluor white (CFW) (**C**), or sodium chloride (NaCl) (**D**) at 5 d post-incubation. (**E**–**G**) Amounts of protoplasts released from WT, deletion mutants, and complementation strains at 30 (**E**), 60 (**F**), and 90 (**G**) min after cell wall-degrading enzymes (CWDE) digestion. (**H**) Images showing protoplasts released from the WT, deletion mutants, and complementation strains at 90 min after CWDE digestion. Bar: 20 µm. (**I**) Transmission electron micrographs showing the morphology of cell walls of the WT and deletion mutants. Bar: 0.2 µm. (**J**) Cell wall thickness of the WT and deletion mutants. (**K**) The accumulation level of GFP-FonSMT3 in Δ*Fonsmt3*-C grown with cell wall-perturbing agents. Δ*Fonsmt3*-C strain was incubated in potato dextrose broth for 36 h and induced by 0.2 mg/mL CR, 0.1 mg/mL CFW, or 0.5 M NaCl for 12 h. Total protein was extracted and subjected to Western blotting, with anti-GFP antibody as a reporter and anti-GAPDH antibody as a control. The experiments were independently performed three times with similar results. Data represented in (**B**–**G**,**J**) are the means ± SD from three independent experiments, and the different letters above the columns denote significant differences at the *p* < 0.05 level.

**Figure 6 jof-09-00094-f006:**
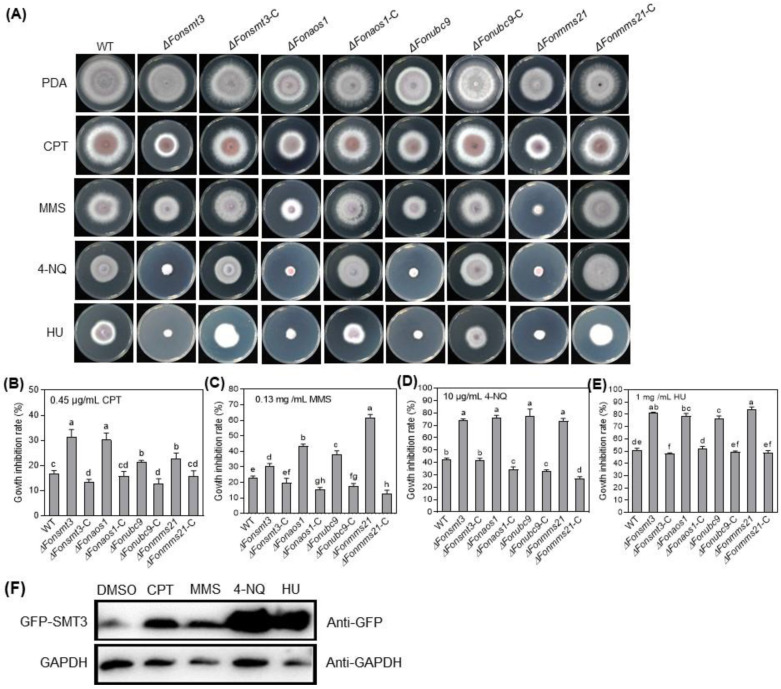
The SUMOylation pathway is involved in *Fusarium oxysporum* f. sp. *niveum* response to DNA damage. (**A**–**E**) Growth phenotype (**A**) and inhibition rate of mycelial growth of the wild type (WT), deletion mutants, and complementation strains grown on potato dextrose agar (PDA) supplemented with camptothecin (CPT) (**B**), methyl methane sulfonate (MMS) (**C**), 4-nitroquinoline (4-NQ) (**D**), or hydroxyurea (HU) (**E**). (**F**) The accumulation level of GFP-FonSMT3 in Δ*Fonsmt3*-C grown with DNA-damaging agents. Δ*Fonsmt3*-C was incubated in potato dextrose broth for 36 h and induced by 0.45 µg/mL CPT, 0.13 mg/mL MMS, 10 µg/mL 4-NQ, or 1 mg/mL HU for 1 h. Total protein was extracted and subjected to Western blotting with anti-GFP antibody as a reporter and anti-GAPDH antibody as a control. The experiments were independently performed three times with similar results. The data shown in (**B**–**E**) are the means ± SD from three independent experiments, and the distinct letters above the columns denote significant differences at the *p* < 0.05 level.

**Figure 7 jof-09-00094-f007:**
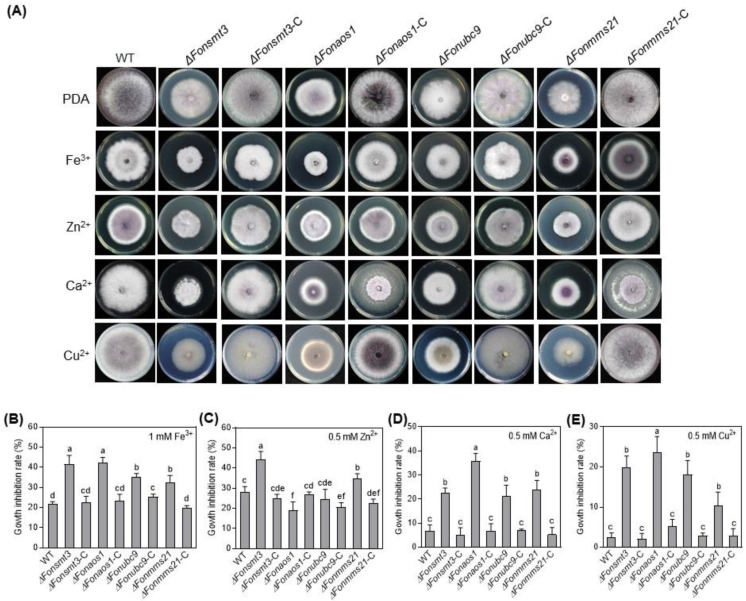
The SUMOylation pathway is involved in metal ions response in *Fusarium oxysporum* f. sp. *niveum*. (**A**–**E**) Growth phenotype (**A**) and inhibition rate of mycelial growth of the wild type (WT), deletion mutants, and complementation strains grown on potato dextrose agar (PDA) supplemented with 1 mM Fe^3+^ (**B**), 0.5 mM Zn^2+^ (**C**), 0.2 M Ca^2+^ (**D**), and 0.5 mM Cu^2+^ (**E**). The experiments were independently performed three times with similar results. The data shown in (**B**–**E**) are the means ± SD from three independent experiments, and the distinct letters above the columns denote significant differences at the *p* < 0.05 level.

**Figure 8 jof-09-00094-f008:**
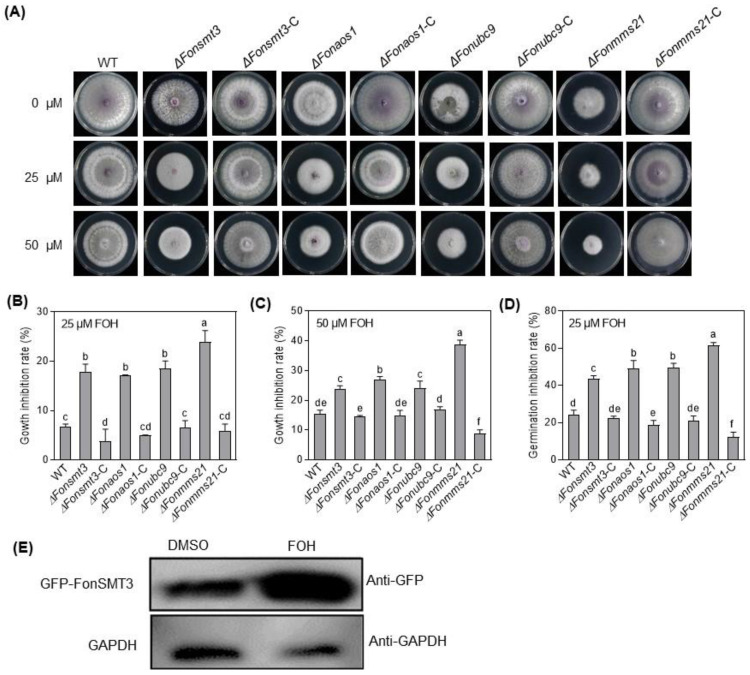
The SUMOylation pathway is involved in apoptosis in *Fusarium oxysporum* f. sp. *niveum*. (**A**–**C**) Growth phenotype (**A**) and inhibition rate of mycelial growth of the wild type (WT), deletion mutants, and complementation strains grown in the absence or presence of 25 (**B**) and 50 (**C**) μM farnesol (FOH). (**D**) Inhibition rate of macroconidia germination of the WT, deletion mutants, and complementation strains in the absence or presence of 25 μM FOH. (**E**) The accumulation level of GFP-FonSMT3 in Δ*Fonsmt3*-C grown with or without 25 μM FOH. Δ*Fonsmt3*-C was incubated in potato dextrose broth for 36 h and induced by 25 μM FOH for 1 h. Total protein was extracted and subjected to Western blotting with anti-GFP antibody as a reporter and anti-GAPDH antibody as a control. The experiments were independently performed three times with similar results. The data shown in (**B**–**D**) are the means ± SD from three independent experiments, and the distinct letters above the columns denote significant differences at the *p* < 0.05 level.

**Figure 9 jof-09-00094-f009:**
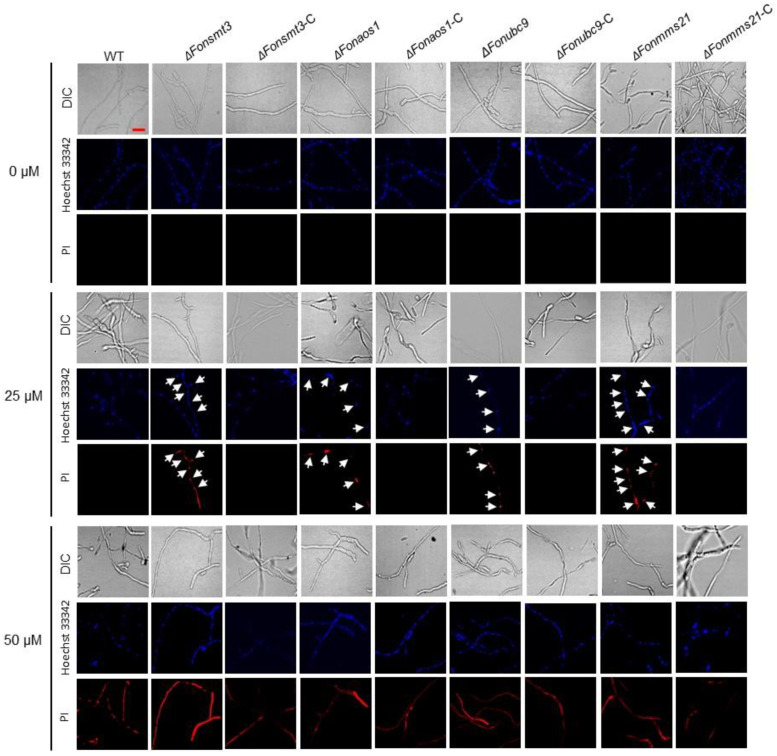
The disruption of the SUMOylation pathway induces apoptotic cell death in *Fusarium oxysporum* f. sp. *niveum*. Macroconidia of the wild type (WT), deletion mutants, and complementation strains were germinated in yeast extract peptone dextrose for 12 h, and the germ tubes were then treated with/without farnesol at 25 or 50 μM for 4 h. Necrotic cells and nuclei were co-stained with propidium iodide (PI) and Hoechst 33342, respectively. Apoptotic chromatin condensation and hyphal necrosis were examined under fluorescence microscopy. Bar: 10 µm. Arrows indicate the chromatin condensation and apoptotic-like cell death in Hoechst 3342 and PI-stained fungal hyphal cells of the deletion mutants, respectively. The experiments were performed three times with similar results.

**Figure 10 jof-09-00094-f010:**
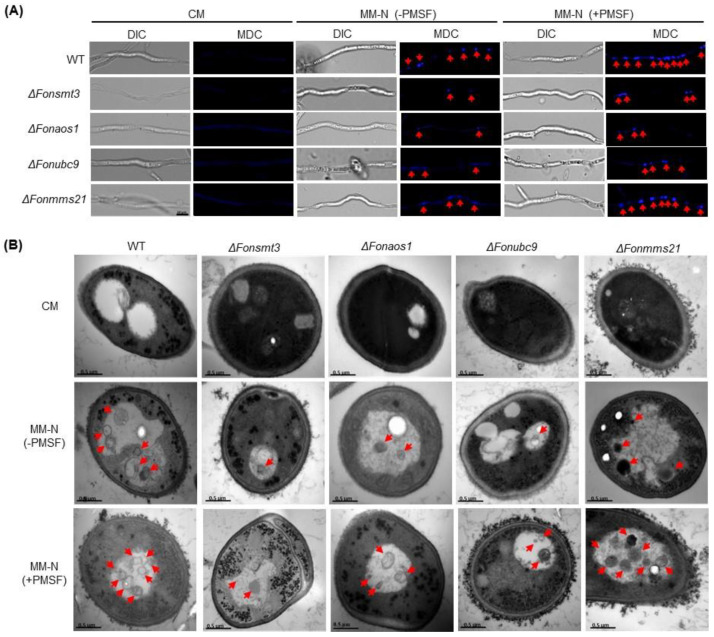
The SUMOylation pathway drives autophagy in *Fusarium oxysporum* f. sp. *niveum*. (**A**) Confocal micrographs displaying hyphae of the wild type (WT) and deletion mutants grown in nitrogen-rich complete medium (CM) or nitrogen-starved minimal medium (MM-N) in the presence (+) or absence (−) of phenylmethanesulfonyl fluoride (PMSF) and stained with monodansylcadaverine (MDC). Bar: 10 µm. (**B**) Transmission electron micrographs of autophagic structures in the mycelial cells of WT and deletion mutants under nitrogen-rich (CM) or nitrogen-starved (MM-N) conditions in the presence (+) or absence (−) of PMSF. Bar: 0.5 µm. Arrows indicate the formation of autophagosomes in mycelial cells (**A**) and the autophagic structures in vacuoles of hyphal cells (**B**) of the tested strains grown under the starvation conditions. The experiments were performed three times with similar results.

## Data Availability

Not applicable.

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
