# Peer review of "The SUMOylation Pathway Components Are Required for Vegetative Growth, Asexual Development, Cytotoxic Responses, and Programmed Cell Death Events in Fusarium oxysporum f. sp. niveum"

_jof, 2023, doi:10.3390/jof9010094_

Round 1

Reviewer 1 Report

The authors identified the SUMOylation pathway components in Fusarium oxysporum f. sp. niveum (Fon). Although some of them seems to be essential genes (deletion of them is lethal), the others were deleted and the deletion mutants were assayed for vegetative growth, asexual reproduction, stress responses, etc. The data presented in the manuscript indicated that the SUMOylation pathway is likely important for the growth and development of Fon. I have few minor comments, mostly related to the phenotype of the complementation strains. If the authors have done repeated experiments, please double-check whether the images in the manuscript represent the real phenotypes. If not, replaced them with better images. If yes, give a very brief description or discuss a little bit (why the defects of the mutants were not fully complemented).  

1 Fig. 2E. The representative images of hyphae (DIC) were quite different. Is that true they have such a big differences. For FonAOS1, are you sure the DIC and Merge images are the same hyphae? 

2 Fig. 3A. Fonaos1-c seems not fully complemented the growth defects (MM culture). 

3 Fig. 3A, Fig. 5A, Fig. 6A and Fig. 7A. The colony of the wild type on PDA is quite different. 

4 Fig. 5A. CFW. Fonubc9-c was quite different from the colony of the wild type and other complementation stains. 

5 Fig. 7A. Ca2+. Fonaos1-c and Fonmms21-c were quite different from the colony of the wild type. 

Author Response

Response to Reviewer 1 Comments

The authors identified the SUMOylation pathway components in Fusarium oxysporum f. sp. niveum (Fon). Although some of them seems to be essential genes (deletion of them is lethal), the others were deleted and the deletion mutants were assayed for vegetative growth, asexual reproduction, stress responses, etc. The data presented in the manuscript indicated that the SUMOylation pathway is likely important for the growth and development of Fon. I have few minor comments, mostly related to the phenotype of the complementation strains.

Response: Thanks for your positive comments. We have revised the manuscript carefully according to your comments; see below for specific responses.

Point 1: If the authors have done repeated experiments, please double-check whether the images in the manuscript represent the real phenotypes. If not, replaced them with better images. If yes, give a very brief description or discuss a little bit (why the defects of the mutants were not fully complemented).

Response: Thanks for your comment. We added some text to discuss this point. (See track changes as shown below, see line no. 372-375 in manuscript file)

Notably, the partial growth or phenotypic recovery of the complementation strains might be due to the non-physiological expression of the genes that were driven by the Rp27 promoter rather than their native promoters.

Point 2: Fig. 2E. The representative images of hyphae (DIC) were quite different. Is that true they have such a big differences. For FonAOS1, are you sure the DIC and Merge images are the same hyphae?

Response: We thank the reviewer for highlighting this point. We added the correct representative image for FonAOS1. (See revised Fig. 2E).

Point 3: Fig. 3A. Fonaos1-c seems not fully complemented the growth defects (MM culture).

Response: Thanks for your comment. We have added some text to discuss this point. (See track changes as shown below; see line no. 372-375 in manuscript file)

Notably, the partial growth or phenotypic recovery of the complementation strains might be due to the non-physiological expression of the genes that were driven by the Rp27 promoter rather than their native promoters.

Point 4: Fig. 3A, Fig. 5A, Fig. 6A and Fig. 7A. The colony of the wild type on PDA is quite different.

Response: We thank the reviewer for highlighting this point. Fig. 5A represents WT phenotype at 5 d post incubation and we have modified the caption to indicate out this point. (See track changes shown below; see line no. 451-454 in manuscript file).

The SUMOylation pathway is involved in maintaining cell wall integrity of Fon. (A-D) Growth phenotype (A) and inhibition rate of mycelial growth of the WT, deletion mutants, and complementation strains when exposed to CR (B), CFW (C), or NaCl (D) at 5 d post incubation.

Point 5: Fig. 5A. CFW. Fonubc9-c was quite different from the colony of the wild type and other complementation stains.

Response: We thank the reviewer for this important point. We have added some text to discuss this point. (See track changes as shown below; see line no. 372-375 in manuscript file)

Notably, the partial growth or phenotypic recovery of the complementation strains might be due to the non-physiological expression of the genes that were driven by the Rp27 promoter rather than their native promoters.

 Point 6: Fig. 7A. Ca2+. Fonaos1-c and Fonmms21-c were quite different from the colony of the wild type.

Response: We thank the reviewer for this important point. We have added some text to discuss this point. (See track changes as shown below; see line no. 372-375 in manuscript file)

Notably, the partial growth or phenotypic recovery of the complementation strains might be due to the non-physiological expression of the genes that were driven by the Rp27 promoter rather than their native promoters.

Reviewer 2 Report

Ullah et al. conducted a thorough molecular study of the SUMOylation pathway components in Fusarium oxysporum f. sp. niveum. While the results of the study are clear, there are a few concerns regarding the writing and presentation of the data that should be addressed. Overall, it is important to ensure that the manuscript is well-written and effectively presents the findings of the study in a clear and concise manner.

Major comments

It is important to avoid repetitive language and to focus on the specific findings of the present study in the manuscript. Instead of consistently stating "The SUMOylation pathway is involved in xxx," it may be more effective to describe the specific role of the SUMOylation pathway in the context of the current study.

In the introduction for each result section, it may be helpful to provide a clear and concise overview of the purpose and significance of the research, and to describe the logical steps that led to the specific hypotheses being tested. It may also be useful to highlight the unique contribution of the present study, rather than simply mentioning findings from other organisms such as Magnaporthe. By providing a clear and compelling narrative, the manuscript can effectively convey the importance and relevance of the research.

Minor comments

Line 44, “limited” is not accurate since many studies have studied SUMO in phytopathogenic fungi.

Line 48 and 84, it may be worthwhile for the authors to check the work of Wang et al. (2022, doi.org/10.1186/s42483-021-00106-w).

Line 86-87, It is important to include and discuss the work of Jian et al. (2022, DOI: 10.1111/nph.18692) in the introduction of the manuscript due to its relevance to the research. It may be helpful to provide a brief summary of the key findings and contributions of this study and to explain how it relates to the present work.

Line 88-89, please provide references to support the statement.

 Fig 1A, presents a decent phylogenetic analysis, but this reviewer would suggest that the authors also provide a similar analysis for the Pfam domain structure of these SUMO proteins to help readers understand the conservation of these proteins. This would make the information clearer and more accessible. Please also label for UQ_con, Ubiquitin, ThiF KYD|A, ThiF, UQ_con, and zf-RING_UBOX in Fig1c.

Line 290, might be essential.

Line 302, Space is needed between “and” and “FonAOS1”.

Line 498, It would be helpful to provide a more detailed description of what is meant by "obvious apoptosis- and necrosis-like cell features."

Line 613, to strengthen the statement, it may be helpful for the authors to consider checking Nagai et al. (2008, DOI: 10.1126/science.1162790) and Huang et al. (2022, DOI: 10.1093/femsre/fuac035).

Line 617, it is clear that Fon is not an outlier of phytopathogenic fungus, so there is no need to consistently mention "including Fon" when discussing this group of organisms. Please check this issue across the manuscript.

Line 660, this reviewer does not find an experiment directly related to virulence.

Author Response

Response to Reviewer 2 Comments

Ullah et al. conducted a thorough molecular study of the SUMOylation pathway components in Fusarium oxysporum f. sp. niveum. While the results of the study are clear, there are a few concerns regarding the writing and presentation of the data that should be addressed. Overall, it is important to ensure that the manuscript is well-written and effectively presents the findings of the study in a clear and concise manner.

Response: Thanks for your comments. We revised the manuscript carefully according to your comments and addressed all questions/concerns in a point-by-point fashion as noted below.

Major comments

Point 1: It is important to avoid repetitive language and to focus on the specific findings of the present study in the manuscript. Instead of consistently stating "The SUMOylation pathway is involved in xxx," it may be more effective to describe the specific role of the SUMOylation pathway in the context of the current study.

Response: We thank the reviewer for this comment. Now, we revised the manuscript to avoid repetition. (See track changes in the manuscript file).

Point 2: In the introduction for each result section, it may be helpful to provide a clear and concise overview of the purpose and significance of the research, and to describe the logical steps that led to the specific hypotheses being tested. It may also be useful to highlight the unique contribution of the present study, rather than simply mentioning findings from other organisms such as Magnaporthe. By providing a clear and compelling narrative, the manuscript can effectively convey the importance and relevance of the research.

Response: As suggested, we modified the introduction section of each result. (See track changes in the manuscript file)

Minor comments

Point 3: Line 44, “limited” is not accurate since many studies have studied SUMO in phytopathogenic fungi.

Response: We modified the sentence. (See track changes as shown below; see line no. 46-47 in manuscript file).

In contrast to other post-translational modifications, such as phosphorylation, methylation and ubiquitination, the SUMOylation process in phytopathogenic fungi is poorly understood.

Point 4: Line 48 and 84, it may be worthwhile for the authors to check the work of Wang et al. (2022, doi.org/10.1186/s42483-021-00106-w).

Response: As suggested, we cited this reference in the text. (See line no. 88 in the manuscript file, ref. [26]).

Point 5: Line 86-87, It is important to include and discuss the work of Jian et al. (2022, DOI: 10.1111/nph.18692) in the introduction of the manuscript due to its relevance to the research. It may be helpful to provide a brief summary of the key findings and contributions of this study and to explain how it relates to the present work.

Response: The suggested modifications have been made. (See track changes as shown below; see line no. 88-89, 91-95 and 644-645 in the manuscript file).

Similarly, the SUMO components deletion led to reductions in the virulence and tolerance to DNA damage agents in Fusarium graminearum [27].

Although the role of SUMO pathway in the stress response and virulence has been established in F. graminearum [27]; however, the basic biological functions, such as growth and development, of the SUMOylation pathway need to be studied in Fusarium species.

This is in line with previous observations describing that the deletion of the SUMOylation pathway components affected the response to DNA damaging agents in M. orayze, F. graminearum, A. nidulans, and A. flavus [9,13,25,27].

Point 6: Line 88-89, please provide references to support the statement.

Response: Thanks for your comment. We added reference to make the statement more clear and understandable. (See track changes; see line no. 91-95 in the manuscript file).

Although the role of SUMO pathway in the stress response and virulence has been established in F. graminearum [27]; however, the basic biological functions, such as growth and development, of the SUMOylation pathway need to be studied in Fusarium species.

Point 7: Fig 1A, presents a decent phylogenetic analysis, but this reviewer would suggest that the authors also provide a similar analysis for the Pfam domain structure of these SUMO proteins to help readers understand the conservation of these proteins. This would make the information clearer and more accessible. Please also label for UQ_con, Ubiquitin, ThiF KYD|A, ThiF, UQ_con, and zf-RING_UBOX in Fig1c.

Response: Thanks for this point. Now, we provided the Multiple Sequence Alignment to indicate out the conservation of Pfam domain structure of SUMO components in different fungal species in the Supplementary Materials as Figure S1-S4.

Point 8: Line 290, might be essential.

Response: Thanks! The suggested modification has been made. (See track changes; see line no. 303 in the manuscript file).

Point 9: Line 302, Space is needed between “and” and “FonAOS1”

Response: Space added. (See track changes; see line no. 318 in the manuscript file)

Point 10: Line 498, It would be helpful to provide a more detailed description of what is meant by "obvious apoptosis- and necrosis-like cell features."more preliminary data to support it.

Response: We added some text to discuss this point. (See track changes as shown below; see line no. 527 in the manuscript file).

At 25 μM FOH level, the germ tubes of the deletion mutants showed the obvious apoptosis- and necrosis-like cell features such as chromatin condensation and cell death.

Point 11: Line 613, to strengthen the statement, it may be helpful for the authors to consider checking Nagai et al. (2008, DOI: 10.1126/science.1162790) and Huang et al. (2022, DOI: 10.1093/femsre/fuac035).

Response: We added some text and provided articles in the discussion part. (See track changes as shown below; see line no. 647-650 in the manuscript file).

The crosstalk between SUMOylation and DNA damage repair in regulating genome stability and virulence in filamentous fungi has been revealed [61,62]; however, in-depth studies are required to elucidate the underlying mechanisms by which SUMO machinery regulates DNA damage response in phytopahtogenic fungi.

Point 12: Line 617, it is clear that Fon is not an outlier of phytopathogenic fungus, so there is no need to consistently mention "including Fon" when discussing this group of organisms. Please check this issue across the manuscript.

Response: Revised as suggested. (See track changes; see line no. 627 and 655 in the manuscript file).

Point 13: Line 660, this reviewer does not find an experiment directly related to virulence.

Response: Thanks for this point. We added some text in the conclusion section to support this statement. (See track changes as shown below; see line no. 694-697 in the manuscript file).

The expression levels of the SUMO components were significantly upregulated during initial infection and gradually decreased in infected watermelon roots at later stages of disease, thus suggesting their potential roles in Fon virulence.